# Integrative Roles of Functional Foods, Microbiotics, Nutrigenetics, and Nutrigenomics in Managing Type 2 Diabetes and Obesity

**DOI:** 10.3390/nu17040608

**Published:** 2025-02-07

**Authors:** Hong Nhung Lam, Shih-Ping Lin, Dang Hien Ngan Nguyen, Chiao-Ming Chen, Chien-Tien Su, Te-Chao Fang, Sing-Chung Li

**Affiliations:** 1School of Nutrition and Health Sciences, College of Nutrition, Taipei Medical University, Taipei 11031, Taiwan; ma07113002@tmu.edu.tw (H.N.L.); ma07112020@tmu.edu.tw (D.H.N.N.); 2Department of Dietetics, Taoyuan Armed Forces General Hospital, Taoyuan 32551, Taiwan; ping06072008@gmail.com; 3Department of Food Science, Nutrition, and Nutraceutical Biotechnology, Shih Chien University, Taipei 10462, Taiwan; charming@g2.usc.edu.tw; 4Department of Family Medicine, Taipei Medical University Hospital, Taipei 11031, Taiwan; ctsu@tmu.edu.tw; 5School of Public Health, College of Public Health, Taipei Medical University, Taipei 11031, Taiwan; 6Division of Nephrology, Department of Internal Medicine, School of Medicine, College of Medicine, Taipei Medical University, Taipei 11031, Taiwan; fangtc@tmu.edu.tw; 7Division of Nephrology, Department of Internal Medicine, Taipei Medical University Hospital, Taipei Medical University, Taipei 11031, Taiwan; 8Research Center of Urology and Kidney, Taipei Medical University, Taipei 11031, Taiwan

**Keywords:** functional foods, microbiotics, nutrigenetics, nutrigenomics, diabetes, obesity

## Abstract

Diabetes and obesity are globally prevalent metabolic disorders posing significant public health challenges. The effective management of these conditions requires integrated and personalized strategies. This study conducted a systematic literature review, identifying 335 relevant papers, with 129 core articles selected after screening for duplicates and irrelevant studies. The focus of the study is on the synergistic roles of functional foods, microbiotics, and nutrigenomics. Functional foods, including phytochemicals (e.g., polyphenols and dietary fibers), zoochemicals (e.g., essential fatty acids), and bioactive compounds from macrofungi, exhibit significant potential in enhancing insulin sensitivity, regulating lipid metabolism, reducing inflammatory responses, and improving antioxidant capacity. Additionally, the critical role of gut microbiota in metabolic health is highlighted, as its interaction with functional foods facilitates the modulation of metabolic pathways. Nutrigenomics, encompassing nutrigenetics and genomics, reveals how genetic variations (e.g., single-nucleotide polymorphisms (SNPs)) influence dietary responses and gene expression, forming a feedback loop between dietary habits, genetic variations, gut microbiota, and metabolic health. This review integrates functional foods, gut microbiota, and genetic insights to propose comprehensive and sustainable personalized nutrition interventions, offering novel perspectives for preventing and managing type 2 diabetes and obesity. Future clinical studies are warranted to validate the long-term efficacy and safety of these strategies.

## 1. Introduction

The escalating prevalence of diabetes and obesity has intensified the search for innovative, integrative strategies for prevention and management. The NCD Risk Factor Collaboration reported that over 1 billion people globally were obese in 2022, with nearly 0.3 billion classified as overweight or obese [1]. The prevalence of obesity among adults in the South-East Asia Region was about 6%, while in the Western Pacific Region, it was approximately 8%. The primary concern with obesity is its strong association with chronic metabolic disorders, including insulin resistance (IR), cardiovascular diseases, and type 2 diabetes (T2DM) [2].

In 2021, an estimated 537 million people globally were living with diabetes, including approximately 206 million adults in the South-East Asia Region (8.7%) and 206 million in the Western Pacific Region (10.9%). These numbers are projected to rise significantly, reaching 643 million globally by 2030 and 783 million by 2045, with a 69% increase anticipated in the South-East Asia Region and similar growth trends in the Western Pacific Region [3]. Beyond conventional pharmaceutical interventions, dietary strategies have emerged as pivotal approaches in health management, emphasizing the roles of functional foods, probiotics, and personalized nutrition guided by advancements in nutrigenomics and nutrigenetics [4]. Functional foods, enriched with bioactive compounds, offer health benefits that extend beyond basic nutrition by modulating physiological functions. Probiotics, as beneficial microorganisms, contribute significantly to gut health and are associated with various systemic health benefits [5]. Personalized nutrition, leveraging insights from nutrigenomics and nutrigenetics, provides tailored dietary recommendations based on individual genetic profiles, recognizing the critical influence of genetic variation on nutrient metabolism and dietary responses [6]. These disciplines explore the intricate interactions between diet, the gut microbiome, and individual genetic profiles, offering insights into metabolic modulation and the potential to mitigate metabolic disorders.

Functional foods, which offer health benefits beyond basic nutrition, include a wide range of bioactive components such as fiber, polyphenols, and omega-3 fatty acids. These foods have been shown to significantly influence the metabolic processes associated with obesity and T2DM by improving insulin sensitivity, regulating blood sugar levels, and modulating fat metabolism [7,8]. Fiber, for example, promotes satiety, regulates blood glucose, and reduces IR, while polyphenols exhibit antioxidant and anti-inflammatory properties that help to mitigate oxidative stress and inflammation—key contributors to metabolic disorders [9,10,11]. Omega-3 fatty acids, found in fatty fish and certain plant oils, have been linked to improved lipid profiles and reduced cardiovascular risks, which are often elevated in individuals with obesity and T2DM [12,13].

Microbiotics, including probiotics, prebiotics, postbiotics, and synbiotics, are essential for maintaining a healthy gut microbiota composition and play a pivotal role in maintaining metabolic and immune homeostasis. Probiotics, live beneficial microorganisms, and prebiotics, non-digestible fibers that support the growth of beneficial microbes, work together to maintain a balanced microbiota critical for optimal gut function, while postbiotics, the byproducts of microbial metabolism, and synbiotics, combinations of probiotics and prebiotics, further enhance gut health by fostering a favorable microbiome [14,15,16,17,18]. The gut microbiota profoundly influences metabolic regulation, affecting processes such as insulin sensitivity, lipid metabolism, and inflammation, dysregulation linked to the progression of diabetic kidney disease (DKD) and chronic kidney disease (CKD) [14,16,19,20]. Dysbiosis contributes to systemic inflammation and the production of gut-derived uremic toxins, exacerbating kidney damage. Studies, such as those by Brugman et al., have demonstrated the critical role of gut microbiota in disease modulation, including the prevention of type 1 diabetes (T1D) onset in a diabetes-prone rat model through antibiotic treatment and dietary interventions [21]. Additionally, gut microbiome imbalances are strongly associated with metabolic disorders such as obesity, type 2 diabetes, and kidney diseases [22,23]. Probiotics, by modulating the gut microbiota, offer a promising therapeutic approach to mitigate these effects through mechanisms such as reducing uremic toxins, enhancing gut barrier integrity, attenuating inflammation, and inhibiting pathogen bacteria growth [24,25]. This review investigates the potential of probiotics as a complementary strategy primarily for managing type 2 diabetes and obesity, while also addressing their secondary benefits in mitigating complications, such as DKD and CKD, emphasizing current evidence and future research directions.

Nutrigenetics and nutrigenomics provide insights into how genetic variations influence dietary responses and how diet modulates gene expression, offering new opportunities for managing complex metabolic disorders such as T2DM and obesity. Key genetic variations, including SNPs in *FTO* (fat mass and obesity-associated) and *PPARγ* (peroxisome proliferator-activated receptor gamma), play significant roles in metabolic regulation, affecting insulin sensitivity, lipid metabolism, and energy balance [26,27,28]. Additionally, host genetics influence gut microbiota composition, which in turn impacts the production of metabolites, such as short-chain fatty acids (SCFAs), critical for metabolic health. The emerging field of nutrigenomics further highlights the interaction between the gut microbiome and the host’s genetic makeup, emphasizing how genetic variations shape individual responses to dietary components and microbiota, affecting susceptibility to metabolic diseases. Integrating genetic and microbiota profiling into personalized nutrition strategies has demonstrated potential in improving metabolic outcomes. Moreover, microbiome modulation—through diet, functional foods, and microbiotic supplementation—is increasingly recognized as a critical mechanism for enhancing metabolic health, offering a promising approach to tailored interventions for preventing and managing obesity and T2DM [29,30].

This review aims to explore the synergistic roles of functional foods, microbiotics, nutrigenomics, and nutrigenetics in modulating the metabolic pathways associated with T2DM and obesity. Figure 1 illustrates the systematic literature review process conducted in this study, which identified 335 relevant papers, with 129 core articles selected after screening for duplicates, non-English, and irrelevant studies. By integrating these approaches, researchers and practitioners can advance holistic and individualized dietary interventions, potentially improving metabolic health and providing sustainable solutions for managing these pervasive conditions.

## 2. Impact of Bioactive Compounds on Metabolic Health in Type 2 Diabetes and Obesity

The role of bioactive compounds in modulating metabolic health has become a focal point in the management of T2DM and obesity. These naturally occurring substances, found in phytochemical, zoochemical, and microchemical forms, offer various mechanisms to influence key metabolic pathways, particularly those involved in insulin sensitivity, fat metabolism, and inflammation. Research into their potential therapeutic effects highlights the importance of bioactive compounds in preventing and managing these chronic diseases (Figure 2).

### 2.1. Phytochemicals

Phytochemicals are natural plant compounds, first introduced by chemist Julius Sachs, with antioxidant, anti-inflammatory, and metabolic-regulating properties, aiding in preventing chronic diseases, like T2DM and obesity, and are key to nutritional interventions [31].

#### Mechanism of Action

Natural compounds have shown significant potential in improving the metabolic dysregulation associated with T2DM and obesity by regulating glucose and lipid metabolism, alleviating oxidative stress, and enhancing insulin sensitivity. Studies indicate that epigallocatechin-3-gallate (EGCG), a major active component of green tea, effectively ameliorates glucose and lipid metabolism while reducing oxidative stress in type 2 diabetic rat models [32]. Dihydro-resveratrol mitigates oxidative stress, adipogenesis, and insulin resistance in high-fat diet-induced obese mouse models and in in vitro systems via AMPK activation [33]. Additionally, the fruits of Rosa laevigata and their bioactive principal sitostenone promote glucose uptake and improve insulin sensitivity in hepatic cells through AMPK/PPAR-γ signaling pathways [34]. Resveratrol, a polyphenolic compound, has garnered significant attention for its clinical applications in managing T2DM and obesity, demonstrating potential in addressing metabolic disorders. However, its limited bioavailability can be overcome through nanotechnology, further enhancing its therapeutic efficacy [35]. Table 1 summarizes the effects of key phytochemicals on metabolic health, highlighting their roles in modulating glucose metabolism, lipid profiles, and oxidative stress. Chronic low-grade inflammation is a key pathological feature of both obesity and T2DM. Phytochemicals, including flavonoids (e.g., quercetin), polyphenols (e.g., curcumin, resveratrol), and carotenoids (e.g., β-carotene), exert potent anti-inflammatory effects by inhibiting NF-κB (nuclear factor kappa B), a major transcription factor involved in the expression of pro-inflammatory cytokines such as tumor necrosis factor-alpha (TNF-α), interleukin-6 (IL-6), and C-reactive protein (CRP). By suppressing NF-κB activation, these phytochemicals reduce the production of inflammatory mediators that promote insulin resistance and β-cell dysfunction, which are key contributors to the development of diabesity [21,36,37,38]. Additionally, certain flavonoids (e.g., quercetin) have been shown to shift macrophage polarization from the pro-inflammatory M1 phenotype to the anti-inflammatory M2 phenotype, further mitigating the inflammatory burden in adipose tissue and improving metabolic health [36,39].

Oxidative stress, primarily driven by the overproduction of reactive oxygen species (ROS), plays a pivotal role in the pathogenesis of obesity and T2DM by impairing insulin signaling and inducing β-cell dysfunction [40,41]. Phytochemicals, such as polyphenols (e.g., resveratrol, curcumin) and carotenoids (e.g., lutein, zeaxanthin), exhibit potent antioxidant properties by activating Nrf2 (nuclear factor erythroid 2-related factor 2), a transcription factor that promotes the expression of endogenous antioxidant enzymes, including superoxide dismutase (SOD), catalase, and glutathione peroxidase. These antioxidants neutralize ROS, thereby reducing oxidative damage to cellular structures, improving insulin signaling, and protecting pancreatic β-cells from oxidative stress [42,43,44]. Consequently, phytochemicals help to maintain glucose homeostasis by improving insulin sensitivity and preventing β-cell dysfunction.

The accumulation of visceral fat is closely associated with insulin resistance and metabolic dysfunction [45]. Several phytochemicals, including omega-3 fatty acids (ALA from flaxseeds) and polyphenols (e.g., resveratrol, curcumin), regulate lipid metabolism by modulating key transcription factors such as PPAR-α and SREBP-1c (sterol regulatory element-binding protein 1c) [46,47,48]. Omega-3 polyunsaturated fatty acids (ω-3 PUFAs) promote fatty acid oxidation in skeletal muscle by activating PPAR-α, thereby improving metabolic regulation, reducing triglyceride levels, and enhancing insulin sensitivity rather than directly influencing HDL (high-density lipoprotein) cholesterol levels [47]. Polyphenols, such as resveratrol, inhibit SREBP-1c, a key regulator of fat synthesis in the liver, thereby reducing hepatic fat accumulation and improving liver function [46,49]. Together, these phytochemicals help to decrease visceral fat, improve lipid metabolism, and enhance fat oxidation, all of which contribute to reducing the risk of obesity-related metabolic diseases like T2DM.

The gut microbiome, including bacteria, archaea, fungi, and viruses, plays a critical role in host metabolism and immune regulation. While probiotics and prebiotics are well-established in gut health, emerging evidence highlights the gut virome’s role in microbial modulation and disease prevention. Probiotics, such as Bifidobacteria, and prebiotics, such as inulin, enhance gut microbiota composition, cognitive function, and metabolic health. Randomized controlled trials demonstrate that their intake improves cognition, reduces body fat, and strengthens gut barrier integrity [50,51,52]. Optimized prebiotic formulations (inulin, FOS, GOS) further enhance probiotic efficacy [53]. Beyond probiotics, the gut virome, particularly bacteriophages, modulates microbial diversity and gut homeostasis. Phages selectively target bacterial populations, influencing obesity, T2DM, and metabolic disorders, while also interacting with immune pathways to regulate inflammation [54]. Integrating probiotics, prebiotics, and microbiotics (phages and virome modulation) offers a holistic strategy for improving gut health and metabolic function. Future research should explore their synergistic potential in precision microbiome-based therapies. Integrating these findings, Table 1 provides a detailed summary of the beneficial effects of phytochemicals on various metabolic pathways, highlighting their multifaceted roles in managing obesity and T2DM.

**Table 1 nutrients-17-00608-t001:** Effects of phytochemicals.

Phytochemical	Species	Experiment Model/Dosage	Key Findings	Reference
Epigallocatechin-3-gallate (EGCG) 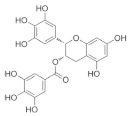	Green tea (*Camellia sinensis*)	-Human intestinal epithelial cells (HIEC) irradiated with up to 8 Gy/2 µM EGCG, 30 min before irradiation.-Male C57 BL/6J mice (9 Gy total body irradiation)/12.5 or 25 mg/kg EGCG for 5 days before and 30 min after radiation.	-Lower γH2AX and 8-OHdG markers in irradiated tissues and cells.-Reduced apoptosis and ferroptosis via ROS scavenging and Nrf2 activation.-Increased GPX4, HO-1, and Slc7A11 expression (dependent on Nrf2).-25 mg/kg EGCG increased mouse survival time from 6.1 to 10.4 days.-Preserved crypt-villus structure, improved villus height, crypt depth, and proliferation (Ki67+ cells).	[55] **
Epigallocatechin gallate (EGCG) 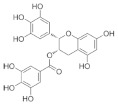 *β*-Cryptoxanthin 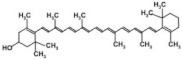	Catechins from green tea (*Camellia sinensis*)*β*-Cryptoxanthin from citrus fruits	Monosodium-glutamate-induced obese male C57BL/6J mice/1.7 mg green tea catechins and 50 µg *β*-Cryptoxanthin/kg/day.	-Reduced body weight in monosodium-glutamate-induced obese mice by approximately 15%.-Decrease in pro-inflammatory M1 macrophages by 30% and an increase in anti-inflammatory M2 macrophages by 25% in adipose tissue.-Serum adiponectin levels increased by 20%.-Reduction in circulating triglycerides by 18% and free fatty acids by 22%.	[56] ***
Epigallocatechin gallate (EGCG) 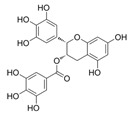 Caffeine 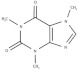	White tea	Obese human participants (BMI ≥ 30 kg/m^2^)/consumed 2 cups of white tea daily (brewed from sachets) along with a calorie-restricted diet (1400–1600 kcal/day) and exercise for 12 weeks *.	-A 15.2% reduction in body weight, a 15.1 cm decrease in waist circumference, and a 5.55-unit drop in BMI.-Inflammatory markers, such as TNF-α, IL-6, and IL-1β, decreased by 21%, 25%, and 20%, respectively.-An 18% reduction in oxidative stress marker MDA and a 22% increase in antioxidant GSH.-Improvements in lipid profiles included a 15% reduction in LDL cholesterol, a 12% decrease in triglycerides, and a 10% increase in HDL cholesterol.	[57] ****
Curcumin 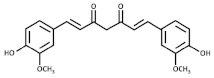	Turmeric (*Curcuma longa*)	3T3-L1 preadipocytes/optimal effects at 10 µM Male C57BL/6J mice with diet induced obesity)/50 mg/kg/day for 8 weeks.	-Improved basal respiration (+16.5%) and ATP production (+27.9%) at 10 µM in vitro.-Increased expression of UCP1 (4.7-fold) and PGC-1α (6.0-fold) in white adipose tissue.-Enhanced mitochondrial function and adipocyte browning via PPARγ activation.	[58] **
Male C57BL/6J mice with diet-induced obesity and genetically obese mice/3% dietary curcumin mixed in the diet for 6 weeks *.	-Reduction in random blood glucose levels up to 20%, HbA1c levels reduced by 15%.-Macrophage infiltration in adipose tissue decreased by 35%, and adiponectin production increased by 25%.-It reduced body fat, increased lean mass, decreased macrophage infiltration in adipose tissue, and lowered hepatic inflammation and NF-κB activity.	[59] ***
Prediabetic human participants (n = 240)/6 capsules per day (1500 mg/day) for 9 months *.	-None of the participants in the curcumin group progressed to type 2 diabetes, compared to 16.4% in the placebo group.-HOMA-β scores (indicating β-cell function) were significantly higher in the curcumin group (61.58 vs. 48.72; p < 0.01).-Adiponectin levels increased by 22%, while HOMA-IR (insulin resistance index) decreased by 20%.	[60] ****
Obese human participants with type 2 diabetes (n = 229)/6 capsules per day (1500 mg/day) for 12 months *.	-Fasting blood glucose reduced significantly in the curcumin group (115.49 mg/dL vs. 130.71 mg/dL in placebo; p < 0.05).-HbA1c levels dropped by 6.12% vs. 6.47%; p < 0.05.-HOMA-β (β-cell function) scores improved by 30%, while leptin levels decreased by 55%.-Adiponectin levels increased by 40%, and HOMA-IR (insulin resistance) decreased by 20%.	[61] ****
Dihydro-Resveratrol (DR2) 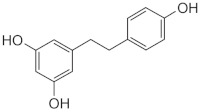	*Dendrobium* spp., *Dioscorea* spp., *Bulbophyllum* spp.	3T3-L1 cells and insulin-resistant HepG2 and C2C12 cells/DR2 (10, 20, 40 µM) for 48 h **.High-fat diet (HFD)-induced obese C57BL/6J mice/DR2 (40 or 80 mg/kg/day) for 3 weeks *.	-Reduced body weight gain, improved glucose tolerance, and enhanced insulin sensitivity.-Decreased adipocyte size, lipid accumulation in adipose tissue and liver, and oxidative stress by activating AMPK and Nrf2 pathways.-Downregulated pro-inflammatory MCP-1 and adipogenic markers (PPAR-γ, C/EBPα, FASN).	[33] **
Resveratrol 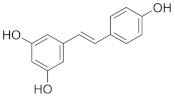	Grapes (*Vitis vinifera*), berries, peanuts (*Arachis hypogaea*).	Patients with type 2 diabetes (n = 110)/200 mg/day (99% pure trans-resveratrol) for 24 weeks.	-Fasting plasma glucose decreased by 7.56%.-HbA1c reduced by 6.31%.-Insulin levels dropped by 9.96%, and HOMA-IR improved by 17.96%.-Inflammatory markers decreased: TNF-α by 13.67%, IL-6 by 13.27%, and hs-CRP by 13.12%.-Oxidative stress marker MDA reduced by 8.46%	[62] ****
Grapes (*Vitis vinifera*), peanuts (*Arachis hypogaea*), red wine.	Obese but otherwise healthy male volunteers (n = 11)/150 mg/day (99% pure trans-resveratrol, resVida™) for 30 days.	-Decreased resting metabolic rate.-Enhanced mitochondrial efficiency and activity through AMPK-SIRT1-PGC-1α pathways.-Plasma glucose, triglycerides, and alanine aminotransferase (ALT) levels decreased significantly.-Systolic blood pressure is reduced by ~5 mmHg.-HOMA-IR (insulin resistance) improved by 25%.-Lowered inflammation markers (IL-6, TNF-α).	[63] ****
Quercetin 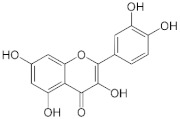	Apples (*Malus domestica*), onions (*Allium cepa*), and berries.	Male C57BL/6J mice were fed a high-fat diet (HF) or HF supplemented with 0.05% quercetin (HFQ) for 6 weeks.	-Reduced obesity biomarkers: body weight by 69.7%, liver weight by 19.6%, and adipose weight by 58.3%.-Inflammatory markers decreased significantly, with plasma insulin reduced by 88%, leptin by 92%, resistin by 27%, and glucagon by 97%.-Metabolic biomarkers improved, including a 25.4% decrease in blood glucose, 62.9% reduction in triglycerides, 37.6% reduction in LDL cholesterol, and an 82.3% improvement in the HOMA-IR index.-Modulated the gut microbiome, reducing the Firmicutes/Bacteroidetes ratio by 65.6%.	[64] ***
Patients with type 2 diabetes mellitus (n = 170)/500 mg/day of quercetin dihydrate for 12 weeks.	-Improved glycemic control, reducing HbA1c levels by up to 4.0% and increasing telomere length by 0.52 kb, suggesting benefits in metabolic health and cellular aging.-A reduction in systolic blood pressure by an average of 6.55 mmHg.	[65,66] ****
*β*-sitosterol 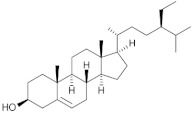	Chia (*Salvia hispanica* L.)	Non-polar fractions (light petroleum and dichloromethane) from the aerial parts of *Salvia hispanica*.	-Antioxidant activity: strong DPPH scavenging (IC50 = 14.73 µg/mL), comparable to ascorbic acid.-Anti-inflammatory activity: effective histamine release inhibition (IC50 = 61.8 µg/mL).-Antidiabetic activity: moderate α-amylase inhibition (IC50 = 673.25 µg/mL).-Anti-obesity activity: significant pancreatic lipase inhibition (IC50 = 59.3 µg/mL).-Cytotoxicity: moderate activity against cancer cell lines (IC50 = 35.9 µg/mL for A-549, IC50 = 42.4 µg/mL for PC-3, IC50 = 47.5 µg/mL for HCT-116).	[67] *
*n*-6 and *n*-3 PUFA 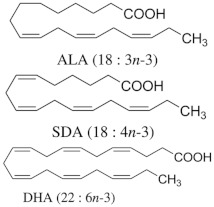	-ALA from linseed oil.-SDA from Echium plantagineum L. oil.-DHA from microalgae oil.	Hypertriglyceridemic adults (n = 59)/ALA group: 20 g/day of linseed oil (7.42 g ALA/day); SDA group: 20 g/day of echium oil (1.57 g SDA/day); DHA group: 12 g/day of microalgae oil (1.64 g DHA/day) for 10 weeks.	-Increased EPA levels in plasma lipids (ALA: +38%, SDA: +73%.-Decreased total cholesterol (ALA: −0.81 mmol/L) and improved LDL/HDL ratio.-Increased DHA levels in plasma and HDL cholesterol by +137% and 10%, respectively.-Decreased n-6/n-3 ratio and improved AA/EPA ratio in SDA and DHA groups.	[68] ****

* in vitro; ** in vitro and in vivo; *** animal model; **** human clinical study. Abbreviation: γH2AX, Phosphorylated H2A histone family member X; 8-OHdG, 8-Hydroxy-2′-deoxyguanosine; ROS, Reactive Oxygen Species; Nrf2, Nuclear Factor Erythroid 2-Related Factor 2; GPX4, Glutathione Peroxidase 4; HO-1, Heme Oxygenase-1; TNF-α, Tumor Necrosis Factor Alpha; IL-6, Interleukin-6; IL-1β, Interleukin-1 Beta; MDA, Malondialdehyde; GSH, Glutathione; LDL, Low-Density Lipoprotein; HDL, High-Density Lipoprotein; UCP1, Uncoupling Protein 1; PGC-1α, Peroxisome Proliferator-Activated Receptor Gamma Coactivator 1-Alpha; ATP, Adenosine Triphosphate; PPARγ, Peroxisome Proliferator-Activated Receptor Gamma; HbA1c, Hemoglobin A1c (Glycated Hemoglobin); NF-κB, Nuclear Factor Kappa B; AMPK, AMP-Activated Protein Kinase; MCP-1, Monocyte Chemoattractant Protein-1 (also known as CCL2); C/EBPα, CCAAT/Enhancer-Binding Protein Alpha; FASN, Fatty Acid Synthase; hs-CRP, High-Sensitivity C-Reactive Protein; AMPK-SIRT1-PGC-1α, AMP-Activated Protein Kinase—Sirtuin 1—Peroxisome Proliferator-Activated Receptor Gamma Coactivator 1-Alpha Pathway; DPPH, 2,2-Diphenyl-1-Picrylhydrazyl (a free radical used in antioxidant assays); EPA, Eicosapentaenoic Acid; ALA, Alpha-Linolenic Acid; SDA, Stearidonic Acid; DHA, Docosahexaenoic Acid.

### 2.2. Zoochemicals

Zoochemicals are natural compounds from animal-based foods, such as omega-3 fatty acids and linoleic acid (CLA), with antioxidant, anti-inflammatory, and metabolic-regulating effects that reduce the risks of cardiovascular diseases and T2DM. The term was introduced by nutrition researchers in the late 20th century to describe health-promoting animal-derived components [69]. Table 2 provides a comprehensive summary of the effects of zoochemicals, highlighting their roles in improving metabolic health and reducing the risks associated with obesity, T2DM, and cardiovascular diseases.

#### Mechanism of Action

Zoochemicals, including omega-3 fatty acids (EPA and DHA), CLA, and milk-derived bioactive peptides, play pivotal roles in improving metabolic health by targeting glucose and lipid metabolism, inflammation, and fat accumulation. Polyunsaturated fatty acids (PUFAs), conjugated linoleic acid (CLA), and bioactive compounds in dairy products play critical roles in metabolic health and inflammation regulation. A balanced dietary ratio of *n*-6:*n*-3 PUFAs has been shown to improve the total fatty acid profile in red blood cells and reduce inflammatory markers, positively impacting inflammation in individuals with obesity [70]. Omega-3 fatty acids were found to reduce the number of adipose tissue macrophages in insulin-resistant subjects, thereby mitigating inflammatory responses [71]. A randomized controlled trial demonstrated that the consumption of goat cheese naturally rich in Omega-3 and CLA improved cardiovascular and inflammatory biomarkers in overweight and obese subjects [72]. Additionally, cis-9, trans-11 conjugated linoleic acid exhibited anti-inflammatory effects in bovine mammary epithelial cells stimulated by Escherichia coli, mediated via inhibition of the NF-κB signaling pathway [73]. A study of Korean adults further revealed that dairy and soy product intake was associated with a reduced 10-year risk of coronary heart disease [74]. Moreover, bioactive peptides derived from goat milk showed potential anticancer effects on the HCT-116 human colorectal carcinoma cell line [75]. As summarized in Table 2, these findings collectively demonstrate that zoochemicals such as PUFAs, CLA, and bioactive components in dairy products hold significant potential in improving metabolic health, reducing inflammation, and lowering the risks of cardiovascular and cancer-related diseases.

Collectively, PUFAs, CLA, and bioactive components in dairy products demonstrate significant potential in improving metabolic health, reducing inflammation, and lowering the risk of cardiovascular and cancer-related diseases.

Chitin and its derivatives (e.g., chitosan, chitosan oligosaccharides, and chitin–glucan fibers) exhibit extensive potential in improving metabolic health by modulating gut microbiota, promoting metabolic signaling, and enhancing therapeutic efficacy. Formulated chitosan microspheres have been shown to remodel gut microbiota and regulate liver miRNA in diet-induced type 2 diabetic rats, improving metabolic functions [76]. According to Lopez-Santamarina et al., insect-based ingredients and insect powder exhibit both beneficial and harmful effects on gene modification regulation, likely due to their high protein content. In contrast, chitin-derived compounds (e.g., chitosan) demonstrate better prebiotic activity in low-protein diets, promoting beneficial bacteria while inhibiting pathogenic microbes. Additionally, chitin derivatives show potential in anti-inflammatory responses, immune stimulation, diabetes prevention, and obesity control. Further research is needed to enhance their application as a dietary fiber source in human nutrition [77].

Carboxymethyl chitin demonstrated anti-obesity effects in 3T3-L1 adipocytes via AMPK and aquaporin-7 signaling pathways [78]. A randomized controlled trial revealed that chitin–glucan fibers reduced oxidized low-density lipoprotein (ox-LDL) levels, indicating potential cardiovascular protective effects [79]. Clinically, chitosan oligosaccharides significantly enhanced the therapeutic efficacy of sitagliptin in Chinese elderly patients with T2DM [80]. Furthermore, dietary chitin–glucan fibers modulated gut bacteria, such as *Roseburia* spp. (Clostridial cluster XIVa), and improved high-fat diet-induced metabolic alterations in mice [81]. These findings highlight the significant potential of chitin and its derivatives in managing T2DM, obesity, and cardiovascular diseases, providing new avenues for functional foods and therapeutic interventions. As summarized in Table 2, these findings collectively demonstrate that zoochemicals such as PUFAs, CLA, bioactive components in dairy products, and chitin and its derivatives hold significant potential in improving metabolic health, reducing inflammation, and lowering the risks of cardiovascular, obesity-related, and cancer-related diseases.

**Table 2 nutrients-17-00608-t002:** Effects of zoochemicals.

Zoochemical	Experiment Model	Dosage and Duration	Key Findings	Reference
Cis-9, Trans-11-Conjugated Linoleic Acid (CLA 9,11) 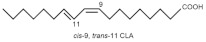	Bovine Mammary Epithelial Cells (BMECs)	600 µmol/L H₂O₂ for 8 h	- Pretreatment with 100 µmol/L c9, t11-CLA significantly increased antioxidant enzyme activities (SOD, CAT, GPx, and T-AOC). - MDA levels decreased from 1.79 to 1.18 in the 100 µmol/L CLA group. - IL-6 and IL-8 levels were significantly decreased in both 50 µmol/L and 100 µmol/L CLA groups. - SOD and GPx1 mRNA expressions were significantly restored in CLA-treated groups. - Pro-inflammatory cytokine mRNA levels (IL-6, IL-8, and IL-1β) were markedly downregulated by CLA treatment.	[82] *
	Alzheimer’s disease (AD) mouse model (hAPPSwInd, J20) and C57BL/6 wild-type mice	0.4% CLA in the diet (~16 mg/day per mouse) from 6 to 14 months of age	- Significantly decreased Aβ40 and Aβ42 levels and reduced hippocampal Aβ deposits. - Increased IL-10 and IL-19 expression by 2.5-fold and 1.8-fold, respectively, and promoted M2 microglial activation. - Increased CLA-lysophosphatidylcholine (CLA-LPC) by ~2-fold. - Increased microglial and astrocytic activity and PPAR-γ activation.	[83] **
Cis-9, Trans-11, and Trans-10, Cis-12 Conjugated Linoleic Acid (CLA 9,11 and CLA 10,12) 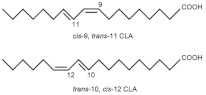	EA.hy926 endothelial cells (human umbilical vein endothelial cell lineage)	1 and 10 µM for CLA9,11 and CLA10,12 for 48 h	- CLA9,11 reduced MCP-1, IL-6, IL-8, and RANTES, while CLA10,12 showed mixed effects, reducing MCP-1 and RANTES but increasing IL-6 levels. - Both CLA isomers decreased COX-2, PPAR-α, and IL-6 gene expression but increased NF-κB1 expression.	[84] *
	Lactating Holstein dairy cows	120 g/day of CLA supplement providing 12 g/day of each isomer from 21 days pre-calving to 60 days post-calving	- Increased milk yield (+3.04 kg/day) and lactose concentration, but decreased milk fat content (−0.62%). - Increased serum glucose (+9.5%, p = 0.01) and insulin levels (p = 0.02), with a trend toward increased IGF-1. - Increased conception rate at first insemination (60% vs. 40%) and elevated progesterone (+28%) and estradiol levels.	[85] **
	Overweight and obese adults (n = 68; BMI ≥ 27 and <40 kg/m^2^)	60 g/day of PUFA-enriched goat cheese for 12 weeks	- Increased HDL cholesterol (+4.5 mg/dL) and the LDL/HDL ratio. - Decreased CRP levels by 36%. - Increased HDL plasma vitamin D.	[72] ***
Polyunsaturated Fatty Acids (PUFAs), including *n*-3 PUFA 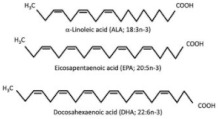 and Conjugated Linoleic Acid (CLA) 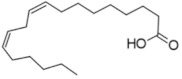	Primary human subcutaneous and visceral adipocytes	100 μM EPA and/or DHA for 72 h	- Reduced expression of inflammatory markers (e.g., IL6, CCL2, CX3CL1) in adipocytes and tissue. - Promoted M2 (anti-inflammatory) over M1 (pro-inflammatory) macrophage differentiation. - Decreased lipid droplet size and lipogenic gene expression (e.g., perilipin A, CIDEA). - DHA had stronger effects on lipogenesis and lipolysis than EPA.	[86] *
Omega-3 fatty acids (EPA and DHA) 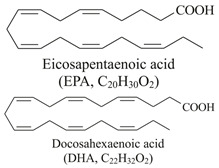	Fish oil such as: Salmo salar (*Atlantic salmon*) Clupea harengus (*Atlantic herring*) Engraulis encrasicolus (*European anchovy*) Sardinops sagax (*Pacific sardine*)	Wistar rats on a high-fat diet (HFD)/3.4% fish oil of total dietary energy for 8 weeks.	- Plasma insulin levels decreased by 39% (1.8 ± 0.2 to 1.1 ± 0.1 ng/mL), and HOMA-IR improved by 39% (3.8 ± 0.4 to 2.3 ± 0.2). - Harmful lipid intermediates, ceramide and diacylglycerol, were reduced by 40% in visceral adipose tissue and by up to 42% in subcutaneous tissue. - Plasma adiponectin increased by 50% (5.2 ± 0.3 to 7.8 ± 0.5 μg/mL), and mitochondrial fatty acid oxidation markers, like CPT1 expression, rose by 60%.	[87] **
		Twelve obese women (BMI ≥ 35) and 12 healthy women (BMI < 24)/4.8 g/day (3.2 g EPA + 1.6 g DHA) for 3 months.	- Insulin levels decreased significantly in the obese group from 13.0 ± 8.8 mU/L to 6.9 ± 3.4 mU/L. - HOMA-IR index decreased by 53% (from 3.2 ± 2.3 to 1.5 ± 0.9). - Reduction in pro-inflammatory TNF-α levels in the obese group.	[88] ***
		60 diabetic patients with NAFLD/2 g/day (180 mg EPA and 120 mg DHA per capsule, 2 capsules daily) for 12 weeks.	- FLI decreased significantly by 3.6 in the omega-3 group compared to 0.9 in the placebo group. - LAP reduced by 14.2 in the omega-3 group versus an increase of 8.0 in the placebo group. - VAI decreased by 0.5 in the omega-3 group compared to 0.0 in the placebo group.	[89] ***
Chitosan 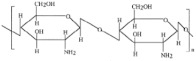	Rats fed with a high-sugar and high-fat diet (HSFD) to induce Type 2 diabetes	Wistar rats on a HSHF/Chitosan microsphere (CMS) supplement providing 40 mg/day for 90 days.	- Remodelled gut microbiota, increasing beneficial bacteria, like Lactobacillus, and reducing pathogenic bacteria. - CMS treatment upregulated miR-203 and downregulated miR-103, modulating glucose and lipid metabolism. - Antidiabetic and anti-inflammatory effects.	[76] **
Carboxymethyl chitin 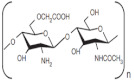	3T3-L1 preadipocytes as a cell model	Carboxymethyl chitin (CM-chitin) was tested in 3T3-L1 adipocytes at concentrations of 50, 100, and 200 μg/mL	- Carboxymethyl chitin activates AMPK signaling. - Aquaporin-7 regulation reduces triglyceride accumulation. - Anti-obesity potential through adipogenesis inhibition.	[78] *
Chitin-glucan complex 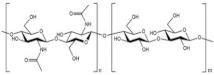	Randomized, double-blind, placebo-controlled clinical trial	Participants received either 1.5 g/day or 4.5 g/day of chitin–glucan for 6 weeks	- Reduction in oxidized LDL levels. - Improvement in lipid profile. - Enhanced antioxidant capacity.	[79] ***

* in vitro; ** animal model; *** human clinical study. Abbreviation: SOD, Superoxide Dismutase; CAT, Catalase; GPx, Glutathione Peroxidase; T-AOC, Total Antioxidant Capacity; Aβ, Amyloid Beta (general term for amyloid beta peptides); MCP-1, Monocyte Chemoattractant Protein-1 (also known as CCL2); RANTES, Regulated upon Activation, Normal T Cell Expressed and Secreted (also known as CCL5); COX-2, Cyclooxygenase-2; NF-κB1, Nuclear Factor Kappa B Subunit 1; IGF-1, Insulin-Like Growth Factor 1; CCL2, C-C Motif Chemokine Ligand 2 (same as MCP-1); CX3CL1, C-X3-C Motif Chemokine Ligand 1 (also known as Fractalkine); CIDEA, Cell Death-Inducing DFFA-Like Effector A; CPT1, Carnitine Palmitoyltransferase 1; FLI, Fatty Liver Index; LAP, Lipid Accumulation Product; VAI, Visceral Adiposity Index.

### 2.3. Macrofungi

In addition to the roles of phytochemical and zoochemical functional components, we have previously reviewed the clinical potential of medicinal components from edible fungi for T2DM treatment and in the prevention of noncommunicable diseases [90]. Macrofungi, including edible and medicinal mushrooms, are emerging as rich sources of bioactive compounds with promising potential for managing metabolic health in T2DM and obesity. These mushrooms are abundant in polysaccharides, terpenoids, phenolic compounds, and sterols, each contributing unique therapeutic properties [91]. For instance, phenolic compounds in *Agaricus bisporus* (white button mushroom) act as potent antioxidants, alleviating oxidative stress, a key factor in IR and β-cell dysfunction. Their antihyperglycemic activity was demonstrated in alloxan-induced diabetic rats, where ethanol (ABEE) and methanol (ABME) extracts reduced serum glucose levels, improved lipid profiles, and restored liver function, with ABEE showing superior efficacy [92]. Similarly, aqueous extracts of *Pleurotus ostreatus* (oyster mushroom) and *Lentinula edodes* (shiitake mushroom) exhibited strong antioxidant, antiviral, and anticancer activities, mediated by bioactive proteins, like superoxide dismutase, and compounds such as catechin and quercetin [93].

Vitamin D plays a role in regulating glucose metabolism and inflammation, potentially reducing the risk of T2DM and obesity. Studies have shown that vitamin D deficiency is associated with insulin resistance, T2DM, and adipose tissue inflammation; appropriate supplementation may improve metabolic health [94]. Hsu et al. first demonstrated that pulsed UV light-treated *Pleurotus citrinopileatus* significantly increased serum 25-hydroxyvitamin D [25(OH)D] levels in healthy adults, addressing vitamin D deficiency. This technique enhances vitamin D_2_ content in mushrooms, providing a safe and sustainable way to boost daily vitamin D intake. It also offers a novel solution for addressing global vitamin D insufficiency and related metabolic issues [95].

Further studies highlight the in vivo therapeutic potential of *Pleurotus ostreatus*-derived insoluble dietary fiber (POIDF) in addressing obesity and metabolic dysregulation. POIDF supplementation in rats reduced body weight, serum lipid levels, and hepatic fat deposition, while enhancing antioxidant capacity and modulating gut microbiota composition. Proteomic and metabolomic analyses revealed its influence on key metabolic pathways, including PPAR and adipocytokine signaling, and increased SCFA production [96]. Additionally, *Lentinula edodes* has been shown to contain novel bioactive compounds with potent antioxidant and anti-inflammatory properties, surpassing the efficacy of standard drugs, like indomethacin, in inhibiting NO and TNF-α production [97].

The submerged cultivation of *Trametes* sp. further emphasizes the potential of mushrooms as functional foods. A phytochemical analysis revealed high levels of saponins, anthraquinones, phenolic compounds, flavonoids, and β-glucan, contributing to strong antioxidant activity and potential applications in managing oxidative stress and metabolic disorders [98]. Evidence from in vitro and in vivo studies supports the therapeutic potential of macrofungi, with their bioactive compounds contributing to anti-inflammatory, antihyperglycemic, and lipid-regulating effects [99,100]. Moreover, a prospective cohort study demonstrated an inverse relationship between mushroom consumption and the risk of T2DM, with higher intakes associated with improved glucose metabolism and reduced oxidative stress. The bioactive compound ergothioneine and other metabolites were identified as significant contributors to these effects. These findings position macrofungi as sustainable and effective dietary components for mitigating inflammation and managing T2DM, particularly when consumed raw or minimally processed to preserve their bioactivity [101].

Hyperglycemia-induced renal damage often triggers inflammation and fibrosis, leading to DKD. Studies on *Cordyceps* species have highlighted their antidiabetic and nephroprotective potential. Fermented *Cordyceps sinensis* (CS) demonstrated an ability to reduce cytotoxicity, inhibit apoptosis, and promote cell proliferation in high-glucose-induced HK-2 cells by modulating key molecular markers, including bax, caspase-3, VEGFA, phosphorylated AKT (P-AKT), and phosphorylated ERK (P-ERK) and PTEN [102]. Similarly, aqueous extracts of *Cordyceps militaris* (CM) in diet streptozotocin-induced diabetic rats significantly lowered blood glucose, improved lipid profiles, and enhanced renal function by reducing albuminuria, serum creatinine, and urea nitrogen levels. CM also attenuated oxidative stress and modulated inflammatory markers, showcasing its potential for managing type 2 diabetes and DKD [103].

## 3. The Potential of Probiotics and Gut Microbiota Modulation in the Treatment of Diabetic Kidney Disease and Chronic Kidney Disease

DKD and CKD are caused by multiple factors, including diabetes, hypertension, chronic inflammation, oxidative stress, and metabolic dysregulation. Persistent hyperglycemia and hypertension lead to glomerular hyperfiltration and high pressure, causing glomerular damage and renal fibrosis. Inflammation and oxidative stress also play critical roles in kidney injury. Dysbiosis of the gut microbiota has been linked to the progression of CKD and DKD, as impaired gut barrier function increases the production of uremic toxins, such as indoxyl sulfate and p-cresyl sulfate, which exacerbate kidney damage. Probiotics, by modulating the gut microbiota, can reduce the generation of harmful metabolites, lower systemic inflammation, and improve renal function, making them a promising approach in the management of CKD and DKD [104,105,106].

*Lactiplantibacillus* plantarum NKK20 has been shown to significantly reduce renal inflammation, serum oxidative stress, and advanced glycation end-product (AGE) levels in diabetic mice, thereby improving kidney damage. Treatment with NKK20 increases the anti-inflammatory metabolite butyrate in feces; metabolomics analysis reveals alterations in 24 metabolites involved in glycerophospholipid and arachidonic acid metabolism. In human renal HK-2 cells, butyrate enhances tight junction gene expression, inhibits fibrosis, and suppresses the PI3K–AKT pathway activation. These findings suggest that NKK20 can effectively prevent and treat diabetic kidney injury by reducing blood glucose levels, decreasing AGE concentrations, and promoting butyrate production [107].

Huang’s study also developed the probiotic *Lactobacillus* mix (Lm), which demonstrated efficacy in improving gut dysbiosis caused by chronic kidney disease (CKD). The probiotic increased short-chain fatty acid production, reduced uremic toxins and related metabolites, alleviated oxidative stress and inflammation, and improved renal function. Both animal and clinical trials revealed that Lm enhanced gut microbiota diversity, reduced toxin accumulation, and mitigated the decline in glomerular filtration rate. Additionally, variable responses in human and feline trials highlighted potential connections between microbial species and metabolites, emphasizing Lm’s precision potential in delaying CKD progression [108].

Vazir et al. highlighted that the gut microbiota functions as a symbiotic ecosystem with both nutritional and protective roles, influenced by the biochemical environment. Their study examined the effects of dietary and pharmacological interventions in uremia and CKD on the gut microbiome. Microbial DNA from the feces of 24 end-stage renal disease (ESRD) patients and 12 healthy individuals was analyzed using phylogenetic microarrays. The results showed significant differences in 190 operational taxonomic units (OTUs) between the ESRD and control groups, with notable increases in OTUs from *Lactobacillus*, *Streptococcus*, *Enterobacteriaceae*, *Halomonadaceae*, *Moraxellaceae*, *Nesterenkonia*, *Polyangiaceae*, *Pseudomonadaceae*, and *Thiotrichaceae* in ESRD patients. A separate study using 5/6 nephrectomized rats revealed significant changes in 175 OTUs, including a marked reduction in *Lactobacillaceae* and *Prevotellaceae*. These findings demonstrate that uremia profoundly alters gut microbiota composition, although the biological implications require further investigation [109].

Kuo et al. demonstrated that approximately one-third of end-stage CKD patients suffer from diabetic nephropathy (DN), which exacerbates renal dysfunction, with few preventive options available. A probiotic combination of *Lactobacillus acidophilus* TYCA06, *Bifidobacterium longum* subsp. *infantis* BLI-02, and *Bifidobacterium bifidum* VDD088 (high dose: 5.125 × 10⁹ CFU/kg/day; low dose: 1.025 × 10⁹ CFU/kg/day) significantly reduced blood urea nitrogen (BUN), serum creatinine, blood glucose, and urinary protein fluctuation rates after 8 weeks in db/db mouse models. The probiotics also improved blood pressure, glucose tolerance, and renal fibrosis. In vitro analysis revealed that TYCA06 and BLI-02 significantly increased acetate production, while all three strains demonstrated antioxidant, anti-inflammatory, and glucose consumption activities. Collectively, this probiotic combination effectively stabilizes glucose levels and slows CKD progression induced by diabetes [110].

## 4. Interplay of Nutrigenetics and Nutrigenomics in Personalized Interventions for Obesity and Type 2 Diabetes

The intersection of nutrigenetics and nutrigenomics represents a transformative approach to understanding the complex relationship between diet, genetic predisposition, and metabolic health, particularly in managing T2DM and obesity. These two complementary disciplines offer profound insights into the genetic and molecular mechanisms underlying individual variability in response to dietary interventions, enabling the development of personalized nutrition strategies tailored to optimize metabolic outcomes [111,112]. Figure 3 illustrates an interactive model of nutrigenetics and nutrigenomics in the management of T2DM and obesity. This model highlights the interconnections among dietary habits, gene variations (e.g., single-nucleotide polymorphisms (SNPs)), gene expression, and gut microbiota, collectively influencing insulin sensitivity, lipid metabolism, and energy balance. Nutritional components in the diet not only modulate gene expression but also reshape the gut microbiota composition, further impacting metabolic functions and providing a scientific basis for personalized nutritional.

To further clarify the distinction between nutrigenetics and nutrigenomics, the latter focuses specifically on how dietary components influence gene expression through epigenetic mechanisms such as DNA methylation, histone modification, and non-coding RNA regulation. These mechanisms reveal how diet interacts with the genome to modulate metabolic health beyond genetic predisposition. For instance, dietary polyphenols, fatty acids, and vitamins have been shown to regulate epigenetic marks, influencing the genes involved in glucose metabolism, lipid homeostasis, and inflammation. By incorporating this expanded understanding of nutrigenomics, this model emphasizes its unique contribution to personalized nutrition through the modulation of gene expression to optimize metabolic outcomes [113,114].

Moreover, according to the study by Scala et al., dietary components, like polyphenols, can activate specific epigenetic pathways, influencing non-coding RNA expression. These mechanisms provide new insights into how diet can reshape gene activity and metabolic health. Mediterranean diet plants, rich in bioactive compounds, such as polyphenols, flavonoids, and terpenoids, have been shown to exert nutrigenomic effects by modulating gene expression through epigenetic mechanisms. Notably, indicaxanthin from prickly pear, kaempferol and quercetin from capers, and terpenoids, like carvacrol and γ-terpinene, from oregano and thyme, exhibit antioxidant, anti-inflammatory, and antimicrobial properties, contributing to metabolic regulation. Additionally, these plants thrive in arid environments, benefiting from plant growth-promoting (PGP) microorganisms that enhance stress resistance and sustainability. This expanded understanding of nutrigenomics underscores its critical role in advancing personalized nutrition strategies by focusing on the modulation of gene expression through epigenetic marks [115].

Research in obesity genetics has significantly advanced our understanding of its monogenic and polygenic forms. Monogenic obesity, often characterized by early-onset and severe obesity, results from rare mutations in genes such as *LEP* (leptin), *MC4R* (melanocortin 4 receptor), and *SH2B1* (SH2B adaptor protein 1) [116,117]. In contrast, polygenic obesity stems from the cumulative effects of numerous genetic variants, each exerting a modest influence on body weight regulation and metabolic processes. Recent genome-wide association studies (GWAS) have identified key loci associated with obesity-related traits, deepening our understanding of the genetic architecture underlying polygenic obesity [118,119]. These advancements lay a critical foundation for the development of personalized strategies aimed at preventing and managing obesity.

Gene–diet interaction studies further emphasize the role of genetic variants in shaping metabolic outcomes. For instance, the Pro12Ala polymorphism of the *PPARγ2* gene was shown to interact with dietary fat intake, where Ala12 carriers exhibited improved insulin sensitivity and reduced BMI when following a Mediterranean diet [120]. Similarly, variants in the *FTO* gene (rs9939609 and rs9930506) were associated with higher BMI and fat intake in Emirati populations, as well as attenuated weight loss responses in Mediterranean diet interventions [121,122].

In the South Indian population, polymorphisms in the *ADIPOQ* gene (e.g., rs2241766 and rs1501299) influenced serum adiponectin levels and conferred differential risks for obesity and T2DM [123]. Additionally, *TCF7L2* gene variants (rs7903146 and rs290487) interacted with BMI and waist circumference to elevate T2DM risk, as shown in Chinese cohorts [124]. In Korean T2DM patients, the rs7903146 T allele of *TCF7L2* was linked to a significantly higher risk of peripheral arterial disease, particularly in those with long-standing diabetes [125].

Furthermore, bioactive compounds from *Hibiscus sabdariffa*, including delphinidin-3-sambubioside (DS3), quercetin (QRC), and hibiscus acid (HA), offer promising insights for nutrigenomics through their interactions with key genes and pathways influencing metabolic health. DS3 primarily targets genes such as *AKT1*, *EGFR*, and *PIK3R1*, modulating the PI3K–AKT signaling pathway. These interactions regulate glucose metabolism, inflammation, and angiogenesis, suggesting DS3’s role in addressing insulin resistance and metabolic dysregulation. QRC affects multiple gene networks, including *CDK2*, *CYP1B1*, and *IGF1R*, involved in metabolic regulation and inflammation. Its impact on the PI3K–AKT pathway and lipid metabolism highlights its potential in personalized strategies for managing obesity and glucose homeostasis. HA uniquely targets genes like *PPARA* and *GABRA2*, influencing neurological pathways such as neuroactive ligand–receptor interactions. These gene interactions suggest potential applications in neurodegenerative disease management and brain health [126].

Izquierdo-Lahuerta’s study demonstrated that the parathyroid hormone-related protein/parathyroid hormone 1 receptor (PTHrP/PTH1R) axis plays a central role in adipose tissue differentiation and remodeling. On the one hand, it is crucial in directing stem cells toward either adipogenesis or osteogenesis. On the other hand, PTHrP/PTH1R appears to be essential in adipose tissue “stress” conditions, whether due to excess fat accumulation in obesity, metabolic syndrome, type 2 diabetes, and gestational diabetes, or disease-induced metabolic dysfunction in cancer and chronic kidney disease [127].

Hernando Boigues et al. reported that PUFAs may regulate obesity-related parameters through epigenetic mechanisms. Studies suggest that PUFAs reversibly alter adipogenesis gene methylation, influencing gene expression and offering the potential for nutritional interventions. Additionally, PUFAs may interact with miRNAs to modulate lipid metabolism, although research on histone modifications remains limited. Current data do not establish an optimal PUFA dosage; however, their role in functional foods and non-pharmacological approaches warrants further study. Given the varying effects of different PUFAs, future research must control for dosage, bioavailability, and genetic backgrounds to clarify their epigenetic impact on obesity [128]. Saad et al. reported that anti-obesity drugs target different adipocyte stages, while polyphenol bioactive compounds (e.g., genistein, apigenin, quercetin, resveratrol) inhibit adipogenesis or induce apoptosis. Phytochemicals regulate epigenetic mechanisms, including DNA methylation, histone acetylation, miRNA, and chromatin remodeling, offering a potential strategy for obesity management. Western diets induce epigenetic alterations; however, natural phytochemicals and nutritional interventions may reverse these effects, benefiting both individual health and future generations’ epigenomes [129].

## 5. Conclusions

This review explored the synergistic roles of functional foods, microbiotics, nutrigenetics, and nutrigenomics in the comprehensive management of T2DM and obesity. The findings demonstrated that bioactive components in functional foods, such as phytochemicals, zoochemicals, and fungal compounds, effectively modulate metabolic pathways, improving insulin sensitivity, lipid metabolism, and inflammatory responses. Microbiotics studies highlighted the critical relationship between gut microbiota and metabolic health, emphasizing the benefits of probiotic and prebiotic supplementation in enhancing gut ecology. Additionally, nutrigenetics and nutrigenomics underscored the influence of genetic variations (e.g., SNPs) on dietary responses and gene expression, enabling personalized nutritional interventions. This research provides comprehensive and sustainable solutions for T2DM and obesity prevention and management, emphasizing the need for further clinical studies to validate the long-term efficacy and safety of these strategies.

## Figures and Tables

**Figure 1 nutrients-17-00608-f001:**
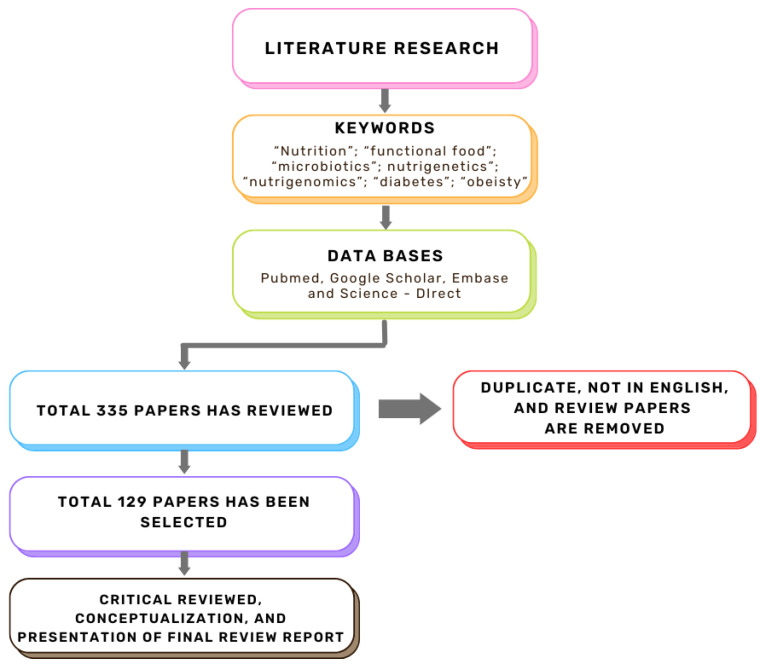
Flow chart of the literature review.

**Figure 2 nutrients-17-00608-f002:**
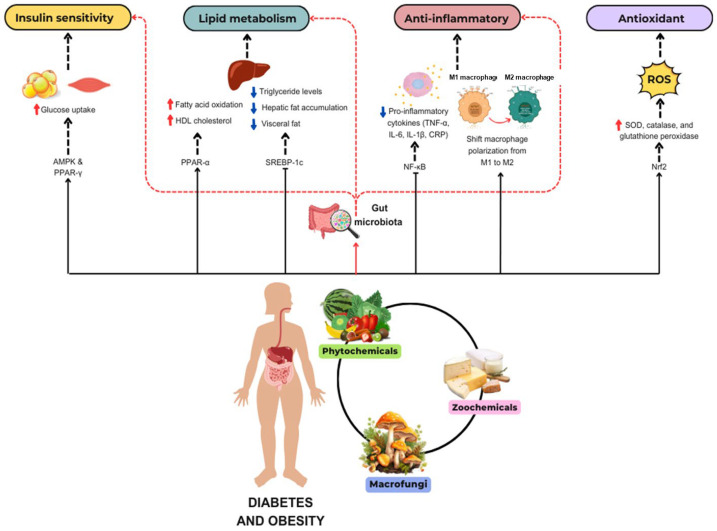
The mechanism of phytochemicals, zoochemicals, and macrofungi on metabolic health in diabetes and obesity. The red arrows indicate activation, blue arrows indicate inhibition, and dotted lines represent indirect effects or interactions.

**Figure 3 nutrients-17-00608-f003:**
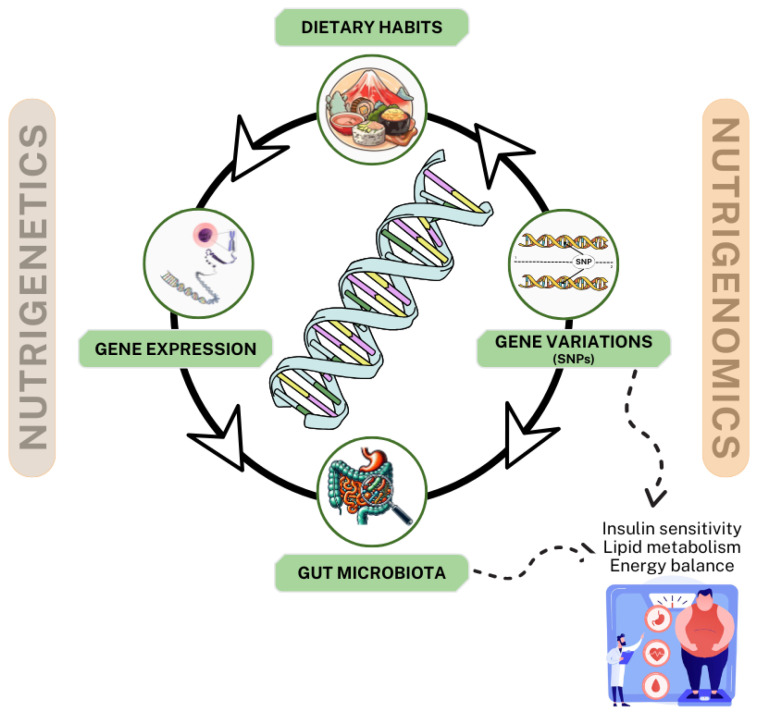
Interactive model of nutrigenetics and nutrigenomics in the management of type 2 diabetes and obesity.

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
