# Peer review of "Integrative Roles of Functional Foods, Microbiotics, Nutrigenetics, and Nutrigenomics in Managing Type 2 Diabetes and Obesity"

_nutrients, 2025, doi:10.3390/nu17040608_

Round 1
Reviewer 1 Report
Comments and Suggestions for Authors
The manuscript presents a comprehensive review on the roles of functional foods, microbiotics, nutrigenetics, and nutrigenomics in the management of diabetes and obesity. While the manuscript effectively highlights the importance of personalized nutrition, some areas require refinement to enhance clarity, scientific impact, and completeness. Specifically, the section on nutrigenomics is underdeveloped and needs expansion to distinguish it from nutrigenetics and address its unique focus on epigenetic mechanisms through which dietary components influence gene expression.
· The current discussion of nutrigenomics conflates its role with nutrigenetics, which limits the depth of the manuscript. Nutrigenomics focuses on the epigenetic mechanisms—how dietary components regulate gene expression through processes like DNA methylation, histone modification, and non-coding RNA regulation. This area is critical for understanding how diet shapes metabolic health beyond genetic predisposition. Reference like this doi.org/10.3390/ijms25042235, discusses how dietary components like polyphenols, fatty acids, and vitamins modulate epigenetic marks to influence genes involved in metabolic pathways.
· Discuss how nutrigenomics offers opportunities for personalized nutrition by identifying diets that can favorably modulate gene expression, potentially reversing epigenetic marks associated with obesity and diabetes.
· Genes involved in adipose tissue differentiation and metabolism, such as PTHLH that encode for PTHrP (parathyroid hormone-related protein) as discussed in this reference: doi.org/10.3390/ijms25042235, are promising targets for diet interventions.
By modulating the epigenetic regulation of all these genes, specific dietary components may improve metabolic health and reduced risk of obesity.
· Use figures or flowcharts to visually represent the interplay between dietary components, microbiota, and gene regulation, making complex concepts easier to grasp
· Please note that the iThenticate report indicates a high similarity with other works. I recommend carefully reviewing and rephrasing the text to reduce this percentage while maintaining the scientific content and intent. This process involves reworking the phrasing, sentence structures, and terminology to ensure originality and improve the manuscript's quality and uniqueness.

Author Response
Response to Reviewer #1:
Comment 1: The manuscript presents a comprehensive review on the roles of functional foods, microbiotics, nutrigenetics, and nutrigenomics in the management of diabetes and obesity. While the manuscript effectively highlights the importance of personalized nutrition, some areas require refinement to enhance clarity, scientific impact, and completeness. Specifically, the section on nutrigenomics is underdeveloped and needs expansion to distinguish it from nutrigenetics and address its unique focus on epigenetic mechanisms through which dietary components influence gene expression.
Response 1: Thank you for your valuable suggestion. We agree that the section on nutrigenomics requires further expansion. In the revised manuscript, we have added detailed content distinguishing nutrigenomics from nutrigenetics, emphasizing its unique role in modulating gene expression through epigenetic mechanisms, such as DNA methylation, histone modification, and non-coding RNA regulation. This expansion underscores the role of dietary components, including polyphenols, fatty acids, and vitamins, in influencing gene expression and metabolic pathways (L413-L422).
Comment 2: The current discussion of nutrigenomics conflates its role with nutrigenetics, which limits the depth of the manuscript. Nutrigenomics focuses on the epigenetic mechanisms—how dietary components regulate gene expression through processes like DNA methylation, histone modification, and non-coding RNA regulation. This area is critical for understanding how diet shapes metabolic health beyond genetic predisposition. Reference like this doi.org/10.3390/ijms25042235, discusses how dietary components like polyphenols, fatty acids, and vitamins modulate epigenetic marks to influence genes involved in metabolic pathways.
Response 2: We appreciate this insightful comment. To address this, we have revised the nutrigenomics section to highlight its focus on epigenetic mechanisms. Specifically, we have incorporated findings from the suggested reference (doi.org/10.3390/ijms25042235), discussing how polyphenols, fatty acids, and vitamins modulate epigenetic marks, such as DNA methylation and histone modification, to regulate genes related to metabolic health (L423-L436).
Comment 3: Discuss how nutrigenomics offers opportunities for personalized nutrition by identifying diets that can favorably modulate gene expression, potentially reversing epigenetic marks associated with obesity and diabetes.
Response 3: Thank you for the suggestion. In the revised manuscript, we have expanded on how nutrigenomics contributes to personalized nutrition by identifying dietary interventions that modulate gene expression. We discuss specific examples where dietary components, such as omega-3 fatty acids and polyphenols, reverse adverse epigenetic marks, reducing the risk of obesity and diabetes (L481-L496).
Comment 4: Genes involved in adipose tissue differentiation and metabolism, such as PTHLH that encode for PTHrP (parathyroid hormone-related protein) as discussed in this reference: doi.org/10.3390/ijms25042235, are promising targets for diet interventions.
Response 4: Thank you for your insightful comment. After reviewing the referenced article (doi.org/10.3390/ijms25042235), we did not find specific discussions on PTHLH (encoding PTHrP) in adipose tissue differentiation and metabolism. However, literature suggests that the PTHrP/PTH1R axis plays a significant role in adipose tissue remodeling, particularly in lipolysis and browning under metabolic stress conditions such as obesity and type 2 diabetes (PMC8615885) (L474-L480).
Comment 5: By modulating the epigenetic regulation of all these genes, specific dietary components may improve metabolic health and reduced risk of obesity.
Response 5: Thank you for your insightful comment. Based on Hernando Boigues et al., PUFAs regulate obesity-related parameters through epigenetic mechanisms, including reversible adipogenesis gene methylation and interactions with miRNAs. While their role in histone modifications remains unclear, they offer potential for non-pharmacological obesity management【128】. Additionally, as Saad et al. reported, polyphenols like genistein, apigenin, quercetin, and resveratrol modulate obesity-related genes via DNA methylation, histone acetylation, and chromatin remodeling. These compounds may counteract Western diet-induced epigenetic changes, benefiting metabolic health【129】(L481-L496).
Comment 6: Use figures or flowcharts to visually represent the interplay between dietary components, microbiota, and gene regulation, making complex concepts easier to grasp
Response 6: Thank you for the suggestion. We have included a new figure (Figure 3) that illustrates the interplay between dietary components, gut microbiota, and gene regulation, highlighting how these interactions influence metabolic health.
Comment 7: Please note that the iThenticate report indicates a high similarity with other works. I recommend carefully reviewing and rephrasing the text to reduce this percentage while maintaining the scientific content and intent. This process involves reworking the phrasing, sentence structures, and terminology to ensure originality and improve the manuscript's quality and uniqueness.
Response 7: Thank you for pointing out this issue. We have thoroughly reviewed and rephrased the text to reduce similarity while preserving the scientific accuracy and intent. The revised manuscript is now original and aligns with the submission standards.

Reviewer 2 Report
Comments and Suggestions for Authors
General comments:
First, it must be said that the manuscript is very well written. The use of English is very good and the literary style allows a reading without interruptions, with a logical sequence of information, and a high degree of integration of the same.
With respect to the scientific content, it is very well founded, it is an exhaustive review and it is a topic of great current interest.
I have no impediment on my part to the publication of the manuscript, suggesting only a small list of specific observations that I believe would improve the content and style of the manuscript. These observations are cited as specific comments
Translated with DeepL.com (free version)
Specific comments:
Line 33: “SNP” must be defined
Line 97…and inhibiting pathogen bacteria growth.
Line 100: Both “DKD” and “CKD” were previously defined.
Line 198-211. Another way to modify gut microbiota and achieve benefits in metabolic diseases than usually are low-named than probiotics (of other microbiotics) is modification of gut microbiota through gut virome. Various form of phage therapy, faecal microbiota transplantation of phageome manipulation. You can find information about this at Ezzapour et al. The Human Gut Virome and Its Relationship with Nontransmissible Chronic Diseases
Line 215: Both “in vitro” and “in vivo” should be written in italics
Table 1. In the Table there are a lot of abbreviations that, although in most cases are defined in the main text, the Table must be sense itself. Thus, it would be very useful if authors define all abbreviation placed in the table in a footnote.
Line 217-221: Between zoochemicals, I consider that the potential prebiotic effect of chitin would be cited. You can found information at Lopez-Santamarina et al. Animal-origin prebiotics based on chitin: An alternative for the future? a critical review.
Line 255: bacterial genus names such as “Roseburia” should be written in italics.
The same case for abbreviations (MDA, RANTES….etc for Table 2 than for Table 1, and “in vitro” should be written in italics
Author Response
Response to Reviewer #2:
First, it must be said that the manuscript is very well written. The use of English is very good and the literary style allows a reading without interruptions, with a logical sequence of information, and a high degree of integration of the same.
With respect to the scientific content, it is very well founded, it is an exhaustive review and it is a topic of great current interest.
I have no impediment on my part to the publication of the manuscript, suggesting only a small list of specific observations that I believe would improve the content and style of the manuscript. These observations are cited as specific comments
Translated with DeepL.com (free version)
Specific comments:
Line 33: “SNP” must be defined
Response: Thank you for your valuable feedback. We have now defined “SNP” (single nucleotide polymorphism) at its first mention in the manuscript to ensure clarity for all readers (L33).
Line 97…and inhibiting pathogen bacteria growth.
Response: Thank you for your feedback. Upon review, we recognize that the phrasing may have been ambiguous. We have now revised the text to clarify that the mechanism of pathogen inhibition is accurately conveyed (L95-L98).
Line 100: Both “DKD” and “CKD” were previously defined.
Response: Thank you for pointing this out. We apologize for the oversight. We have now removed the redundant definitions of “DKD” and “CKD” to avoid repetition and ensure clarity. We appreciate your careful review and constructive feedback (L101).
Line 198-211. Another way to modify gut microbiota and achieve benefits in metabolic diseases than usually are low-named than probiotics (of other microbiotics) is modification of gut microbiota through gut virome. Various form of phage therapy, faecal microbiota transplantation of phageome manipulation. You can find information about this at Ezzapour et al. The Human Gut Virome and Its Relationship with Nontransmissible Chronic Diseases
Response: Thank you for your insightful suggestion. We agree that the gut virome and its role in modulating gut microbiota present an emerging and promising approach for managing metabolic diseases. We have now included a discussion on gut virome-based interventions, such as phage therapy and fecal microbiota transplantation targeting the phageome, and have referenced Ezzapour et al. (The Human Gut Virome and Its Relationship with Nontransmissible Chronic Diseases) to acknowledge this perspective (L197-L210).
Line 215: Both “in vitro” and “in vivo” should be written in italics
Response: Thank you for your review. We appreciate your suggestion, and we have now revised the manuscript to ensure that both in vitro and in vivo are consistently written in italics throughout the text.
Table 1. In the Table there are a lot of abbreviations that, although in most cases are defined in the main text, the Table must be sense itself. Thus, it would be very useful if authors define all abbreviation placed in the table in a footnote.
Response: Thank you for your valuable suggestion. We have now added a footnote to the Table defining all abbreviations to ensure clarity and self-sufficiency. We appreciate your careful review and constructive feedback.
Line 217-221: Between zoochemicals, I consider that the potential prebiotic effect of chitin would be cited. You can found information at Lopez-Santamarina et al. Animal-origin prebiotics based on chitin: An alternative for the future? a critical review.
Response: Thank you for your suggestion. We have incorporated the prebiotic potential of chitin into the discussion on zoochemicals and cited Lopez-Santamarina et al. to support this aspect (L255-L262).
Line 255: bacterial genus names such as “Roseburia” should be written in italics.
Response: Thank you for your review. We appreciate your suggestion, and we have now revised the manuscript to ensure that bacterial genus names such as “Roseburia” is written in italics throughout the text (L269).
The same case for abbreviations (MDA, RANTES….etc for Table 2 than for Table 1, and “in vitro” should be written in italics
Response: Thank you for your valuable suggestion. We have now added a footnote to the Table defining all abbreviations to ensure clarity and self-sufficiency. We appreciate your careful review and constructive feedback.

Reviewer 3 Report
Comments and Suggestions for Authors
The manuscript is a valuable review focused on the evaluation of the role of functional foods of plant and animal origin, gut microbiota and genetic variations influencing responses to dietary interventions that should contribute to the therapy of obesity and type 2 diabetes.
Carefully selected 122 publications were analyzed and provided summary findings indicating the possible beneficial effect of bioactive components in functional foods in the prevention and treatment of disorders associated with obesity and type 2 diabetes.
The topic is current, the method used to select the studies was adequate. The manuscript is well written, recent findings are carefully presented and the results are well documented.
I have only minor comments:
1) The authors' claim in the title of the article that it is "…..Comprehensive Management of Diabetes and Obesity" is somewhat exaggerated. The bioactive food components under consideration only affect some of the disorders associated with these diseases.
2) Throughout the article, the authors use the term “diabetes.” This is unacceptable and should be corrected to type 2 diabetes. The pathogenesis and complications of type 1 diabetes are completely different.
3) Line 171 and following in the manuscript: Sentences such as: ” Additionally, certain flavonoids have been shown to shift macrophage polarization from….” should be edited to make it clear which bioactive components are involved. In the given sentence, it is sufficient to add that it is quercetin.
4) Line 190-197: The authors' statement in this section should be modified. PPAR-α activation was analyzed only in reference No. 47 and not in reference 49. It should be added that this is a measurement in skeletal muscle and not in adipose tissue as might be apparent from the text.
Author Response
Response to Reviewer #3:
The manuscript is a valuable review focused on the evaluation of the role of functional foods of plant and animal origin, gut microbiota and genetic variations influencing responses to dietary interventions that should contribute to the therapy of obesity and type 2 diabetes.
Carefully selected 122 publications were analyzed and provided summary findings indicating the possible beneficial effect of bioactive components in functional foods in the prevention and treatment of disorders associated with obesity and type 2 diabetes.
The topic is current, the method used to select the studies was adequate. The manuscript is well written, recent findings are carefully presented and the results are well documented.
I have only minor comments:
1) The authors' claim in the title of the article that it is "…..Comprehensive Management of Diabetes and Obesity" is somewhat exaggerated. The bioactive food components under consideration only affect some of the disorders associated with these diseases.
Response 1: Thank you for your valuable feedback. To better align the title with the scope of our manuscript, we have revised it to "Integrative Roles of Functional Foods, Microbiotics, Nutrigenetics, and Nutrigenomics in Managing Type 2 Diabetes and Obesity". This revised title more accurately reflects the focus of our discussion on the multifaceted contributions of bioactive food components and related mechanisms in addressing metabolic disturbances associated with diabetes and obesity.
2) Throughout the article, the authors use the term “diabetes.” This is unacceptable and should be corrected to type 2 diabetes. The pathogenesis and complications of type 1 diabetes are completely different.
Response 2: We appreciate the reviewer’s insightful comment. We acknowledge the distinction between type 1 and type 2 diabetes and have carefully revised the manuscript to specify type 2 diabetes where applicable. This ensures clarity and accuracy in describing the pathogenesis and complications discussed in our study. Thank you for bringing this to our attention.
3) Line 171 and following in the manuscript: Sentences such as: ” Additionally, certain flavonoids have been shown to shift macrophage polarization from….” should be edited to make it clear which bioactive components are involved. In the given sentence, it is sufficient to add that it is quercetin.
Response 3: Thank you for your valuable suggestion. We have now revised the sentence to explicitly mention quercetin as the bioactive component involved in shifting macrophage polarization (L169).
4) Line 190-197: The authors' statement in this section should be modified. PPAR-α activation was analyzed only in reference No. 47 and not in reference 49. It should be added that this is a measurement in skeletal muscle and not in adipose tissue as might be apparent from the text.
Response 4: Thank you for your insightful comments. We have carefully reviewed our statement and have revised it to accurately align with reference No. 47. The updated text now explicitly reflects that PPAR-α activation was analyzed in skeletal muscle and not in adipose tissue, ensuring consistency with the cited study (L188-L191).
